# Effect of mRNA Delivery Modality and Formulation on Cutaneous mRNA Distribution and Downstream eGFP Expression

**DOI:** 10.3390/pharmaceutics14010151

**Published:** 2022-01-08

**Authors:** Aditya R. Darade, Maria Lapteva, Thomas Hoffmann, Markus Mandler, Achim Schneeberger, Yogeshvar N. Kalia

**Affiliations:** 1School of Pharmaceutical Sciences, University of Geneva, CMU, 1 Rue Michel-Servet, 1211 Geneva, Switzerland; Aditya.Darade@unige.ch (A.R.D.); Maria.Lapteva@unige.ch (M.L.); 2Institute of Pharmaceutical Sciences Western Switzerland, University of Geneva, 1211 Geneva, Switzerland; 3Accanis Biotech F&E GmbH & Co KG, Vienna Biocenter, Karl-Farkas-Gasse 22/OG 4, 1030 Vienna, Austria; Thomas.Hoffmann@accanis.com (T.H.); Markus.Mandler@accanis.com (M.M.); Achim.Schneeberger@accanis.com (A.S.); 4Tridem Bioscience GmbH & Co KG, Vienna Biocenter, Dr.-Bohr-Gasse 7, 1030 Vienna, Austria

**Keywords:** mRNA delivery, biodistribution, intradermal injection, jet injection, microneedle injection, fractional laser ablation, eGFP, formulation, visualization, protein expression

## Abstract

In vitro transcribed messenger ribonucleic acid (mRNA) constitutes an emerging therapeutic class with several clinical applications. This study presents a systematic comparison of different technologies—intradermal injection, microneedle injection, jet injection, and fractional laser ablation—for the topical cutaneous delivery of mRNA. Delivery of Cy5 labeled mRNA and non-labeled enhanced green fluorescent protein (eGFP) expressing mRNA was investigated in a viable ex vivo porcine skin model and monitored for 48 h. Forty 10 µm-thick horizontal sections were prepared from each skin sample and Cy5 labeled mRNA or eGFP expression visualized as a function of depth by confocal laser scanning microscopy and immunohistochemistry. A pixel-based method was used to create a semi-quantitative biodistribution profile. Different spatial distributions of Cy5 labeled mRNA and eGFP expression were observed, depending on the delivery modality; localization of eGFP expression pointed to the cells responsible. Delivery efficiencies and knowledge of delivery sites can facilitate development of efficient, targeted mRNA-based therapeutics.

## 1. Introduction

The administration of protein-based therapeutic agents to treat dermatological diseases has brought great relief to patients suffering from severe skin conditions, such as plaque psoriasis, atopic dermatitis, or skin cancer [1,2,3]. However, their physicochemical properties and susceptibility to degradation necessitate that the administration of such therapeutics must be parenteral, even if the disease is localized, leading to the risk of numerous and serious systemic off-target side effects and limiting more widespread use [3,4]. 

Although topical administration of therapeutic proteins to skin is appealing, it is hindered by their physiochemical properties (e.g., high molecular weight) and inability to cross the skin barrier. Several reports show the feasibility of delivering biotherapeutics to skin using minimally invasive [5,6] or non-invasive methods [7,8,9]. However, only in vitro experiments were conducted, and the skin residence time of these drugs remains unknown. A prolonged and targeted presence of therapeutic proteins at the diseased site would be of considerable interest and could be brought about by the therapeutic administration of nucleic acid-based agents, commonly known as “gene therapy”. It consists of the delivery of vectorized genetic material to “modify or manipulate the expression of a gene or to alter the biological properties of living cells for therapeutic use” [10] and, thus, enables the local expression of therapeutic proteins. The expressed proteins can be endogenous but deficient or exogenous but directed at a specific disease target; they can act locally at the injection site or be distributed to the whole organism.

Initially, the delivered material was DNA-based: the first product to have received regulatory approval in 2012 was Glybera^®^, a human lipoprotein lipase gene loaded into an adeno-associated virus serotype 1 (AAV1) viral vector delivered to muscle cells to treat lipoprotein lipase deficiency. It has recently been withdrawn from the European market: the approval renewal was not filed by Unique due to the rarity of the disease and increased regulatory requirements [11]. However, the number of FDA and EMA approved gene therapies for the treatment of different protein deficiencies has rapidly expanded: Strimvelis^®^ [12], Luxturna^®^ [12,13], and Zolgensma^®^ [14]. While DNA is commonly used for gene therapy, it has many disadvantages, including the necessity for the DNA to cross not only the cell membrane but also the nuclear membrane barrier, a particularly difficult challenge in an in vivo environment [15]. Furthermore, the possibility of mutagenesis due to integration of the material into the genome increases safety concerns [16]. These issues combine to make DNA-based therapies challenging.

In contrast, mRNA-based approaches are becoming increasingly popular. Indeed, the intrinsic attributes of mRNA render it quite advantageous as a genetic medicine: for example, mRNA is translated into a protein in the cytoplasm, which makes its expression independent of entry into the nucleus or cell cycle. Furthermore, mRNA does not integrate into the host genome and only expresses temporarily, which minimizes potential side effects and allows a more flexible treatment [17].

In vitro transcribed mRNA (IVT-mRNA) can be engineered to express specific proteins by structurally resembling natural mRNA. The IVT-mRNA has to enter the cytoplasm from the extracellular space, in contrast to natural mRNA that is produced in the nucleus and enters the cytoplasm through nuclear export [18]. Even the small amounts of therapeutic proteins encoded by the IVT-mRNA are sufficient to exert a significant response (except for the protein replacement therapies).

However, the challenges posed by mRNA stability and delivery have remained as two significant obstacles to mRNA therapy. Therefore, the need for suitable delivery methods that can efficiently deliver mRNA has increased significantly [19,20,21,22,23]. An ideal mRNA delivery system will protect it from degradation by nucleases, facilitate cellular uptake, and endosomal escape, demonstrate appropriate stability, efficacy, and biodistribution in vivo along with low levels of toxicity and immunogenic response: mRNA delivery strategies can be differentiated into three categories, employing physical, biological, or chemical methodologies [19]. Electroporation has probably been the most commonly used “physical” method to deliver mRNA [20]. Biological approaches include the use of naturally existing viruses, such as adenoviruses, adeno-associated viruses, and lentiviruses, as vectors for mRNA delivery. Chemical approaches to deliver mRNA involve the use of materials, such as lipids and polymers. These materials can complex with mRNA into liposomes or nanoparticles and can be used as non-viral vectors for its delivery. So far, lipids and polymer-based materials are the major non-viral mRNA delivery systems. Typically, cationic lipids and polymers are used to form complexes based on electrostatic and hydrophobic interactions with the negatively-charged mRNA [21,22,23]. 

While some small interfering RNA (siRNA)-based therapeutics have been approved by the US FDA and EMA and are available in the market for the treatment of rare hereditary conditions: Onpattro^®^ [24], Tegsedi^®^ [25], Spinraza^®^ [26,27], few mRNA-based therapies have yet reached the market. Various mRNA-based vaccines were quickly developed and approved to counter the novel coronavirus (SARS-CoV-2) pandemic. The first product, Comirnaty^®^ (approved on 21st December 2020 in EU) was developed by BioNTech and Pfizer. It consists of mRNA encoding for the viral spike (S) protein of SARS-CoV-2 formulated into lipid nanoparticles [28]. The vaccine, administered intramuscularly, was found to be 95% effective in preventing SARS-CoV-2 (COVID-19) in Phase 3 clinical trials [29,30]. A concurrent product, Spikevax^®^, was successfully brought to market one month later by Moderna Therapeutics. Their mRNA-1273 vaccine also consists of mRNA encoding for the viral spike (S) protein of SARS-CoV-2 formulated into lipid nanoparticles [28]. Moderna Therapeutics has many other mRNA therapeutic products in the pipeline, including mRNA-4157, which is a personalized cancer vaccine in combination with Merck’s pembrolizumab (Keytruda^®^). It is currently in Phase 2 clinical trials, expected to be finished by 2023 [31,32]. 

mRNA AZD-8601 is an investigational mRNA-based therapy being developed by AstraZeneca in collaboration with Moderna Therapeutics that encodes for vascular endothelial growth factor-A (VEGF-A). When injected intradermally, VEGF-A mRNA may potentially lead to the formation of more blood vessels and, hence, improve blood supply. mRNA AZD-8601 could be used for regenerative treatments for patients with heart failure or after a heart attack, or diabetic wound healing, as well as other ischemic vascular diseases. Gan et al. reported that upon intradermal injection of naked AZD-8601 in men with type 2 diabetes mellitus, VEGF-A was produced in the skin and enhanced the basal skin blood flow at 4 h and 7 days, which was confirmed using laser Doppler fluximetry and imaging [33].

It is paramount to note that the aforementioned mRNA therapeutics were injected to allow a localized protein expression – either to induce an immune response against an antigen in the muscle in the case of the vaccines or to exert a therapeutic action on the disease target, in the dermis, in the case of blood flow improvement. This ability to allow targeted local protein expression is of major interest for dermatological diseases. For example, Hochmann et al. investigated the biolistic delivery (gene gun) of INFα and eGFP expressing mRNA coated on gold particles to human skin ex vivo for the treatment of non-melanoma skin cancer. Gold particles were principally found in the epidermis and to a lesser extent in the dermis: eGFP expression was located in the upper layers of the epidermis and for INFα in the whole epidermis [34]. In light of these findings, it was of interest to know whether or not dermal delivery of mRNA would be needed in order to have dermal protein expression. 

Consequently, the main aim of the present study was to improve our understanding—both qualitatively and quantitatively—of mRNA distribution and expression in the skin after formulation application using different delivery techniques. To address these questions, we delivered Cy5 labeled mRNA and unlabeled eGFP expressing mRNA to viable porcine skin ex vivo by using different delivery techniques targeting different depths and hence anatomical regions of the skin. Fluorophore tagged mRNA enabled visualization of mRNA in the skin and helped to understand distribution pathways, whereas eGFP expression provided insight into the extent and sites of expression in the different skin layers. 

The specific objectives of the present study were: (i) to develop robust methods for the visualization of mRNA delivery and subsequent protein expression, (ii) to compare different delivery techniques, including invasive methods (conventional intradermal injection and hollow microneedle (MicronJet™600, NanoPass Technologies Ltd., see below) intradermal injection), “less-invasive” methods (Dermojet^®^ injection system, AKRA DERMOJET), and minimally-invasive methods (P.L.E.A.S.E.^®^ Er:YAG fractional laser ablation; Pantec Biosolutions AG), in terms of mRNA delivery and protein expression, (iii) to elucidate the effect of different stabilizers on mRNA delivery and protein expression, (iv) to develop a semi-quantitative technique to evaluate mRNA delivery and subsequent protein expression. 

## 2. Materials and Methods

### 2.1. Materials

Hydrogen peroxide (30% *v*/*v*), TRIS base, Goat serum, formaldehyde solution, Glycerol, Mowiol^®^ 8-88 poly(vinyl alcohol), Eukitt mounting medium, and PBS were purchased from Sigma-Aldrich (Steinheim, Germany). DAPI (4′,6-diamidino-2-phénylindole) and Tween 20 were purchased from Applichem Axon lab AG (Baden-Dättwil, Switzerland). The primary antibody: mouse anti-GFP, clone 9F9.F9, mIgG1 was provided by Abcam (Cambridge, UK). The secondary antibody: Alexa Fluor™ 594 goat anti-mouse IgG [H + L] was provided by Thermofisher Scientific, Life technologies (Plan-les-Ouates, Switzerland). Acetone was purchased from Fisher Scientific (Reinach, Switzerland). All other reagents were at least reagent grade. All other solvents were HPLC grade (HiPerSolv Chromatonorm; Darmstadt, Germany). PBST was used for some washing procedures (0.05% Tween 20 in PBS), Ultra-pure water (Millipore Milli-Q Gard 1 Purification Pack resistivity >18 MΩ·cm; Zug, Switzerland) was used to prepare all solutions.

### 2.2. Procuring and Processing Viable Porcine Skin

The experiments were performed on freshly excised full-thickness porcine skin (abdominal region) obtained from a local abattoir (Fleischerei Nötsch, Puchberg, Austria). The skin surface was cleaned with a hydroalcoholic solution to reduce the bioburden and to avoid interference from surface ribonuclease.

### 2.3. Experimental Conditions

#### 2.3.1. mRNA

The mRNAs (TriLink, San Diego, CA, USA) under investigation were administered to the viable porcine skin using various delivery systems. Two different mRNAs were used: (i) mRNA labeled with Cy5 (CleanCap™ Cyanine 5 FLuc mRNA (5 moU)—TriLink Biotechnologies, tebu-bio GmbH, Offenbach am Main, Germany) enabled visualization of the distribution of mRNA in the skin using widefield/confocal fluorescent microscopy and (ii) eGFP expressing mRNA (CleanCap™ eGFP mRNA (5 moU)—TriLink Biotechnologies, tebu-bio GmbH, Offenbach am Main, Germany), which reported on whether the mRNA delivered was expressed in the skin. Expression of eGFP post-administration of mRNA was visualized by fluorescence microscopy. Before administration, mRNAs were mixed with transfection agents. Two different transfection agents were studied; a polymer-based transfection agent (in vivo jet-PEI^®^; Polyplus transfection, Illkirch, France) and a liposomal system (jetMESSENGER™; Polyplus transfection, Illkirch, France) made of ionizable mono-cationic lipids and co-helper phospholipids [35]. 

#### 2.3.2. mRNA Delivery Techniques

To deliver the mRNA, four different systems targeting different regions of skin were studied. The experimental variables and delivery systems are summarized in Table 1.
**Intradermal injections**. Intradermal (ID) injection and hollow microneedle intradermal injection systems (MicronJet™600) were selected among “invasive” techniques. A 26 G × 3/8″ needle was used to inject mRNA formulations into the dermal region of porcine skin. Although ID injection is a commonly used technique for the targeted delivery of actives into the dermis (e.g., BCG and other vaccines; Mendel–Mantoux tuberculin test; local anesthesia and aesthetic surgery) [36], it requires specific training – for example, the injection angle must be low (10–15°) and the injection volumes should be < 0.5 mL. Another ID injection-based delivery system tested was the FDA- and EMA-approved MicronJet™600 (NanoPass Technologies, Ltd., Nes Ziona, Israel). This consists of an array of three 600 μm microneedles, made from silicon, that can be easily mounted on a syringe. The microneedles are 600 μm in length and already beveled to facilitate insertion into the skin at the correct angle [37]. This system has already been proven effective for mRNA delivery [38]. **Jet injector**. A needle-free jet injector system, Dermojet^®^ developed by AKRA DERMOJET (Pau, France), was also used to deliver mRNA formulations. Liquid-jet injectors use compressed gas or spring to generate a high-velocity jet (with velocities ranging from 100 to 200 m/s) propelled from a nozzle with a pressure of about 1420 psi. Depending on the jet velocity and orifice diameter, the jet can be delivered into the dermis or deeper [39]. Successful drug delivery and good clinical outcomes have already been achieved using this device [40]. mRNA delivery was investigated using 2 different “heads”: one with a single nozzle and the other with a triple nozzle, i.e., enabling simultaneous injection at three different sites.**Fractional laser ablation**. Minimally invasive fractional laser ablation has also been demonstrated as being able to deliver macromolecules [41,42,43,44] and positive clinical outcomes have been reported for the laser assisted delivery of etanercept for the treatment of psoriasis in a phase 1 study [45]. Low-intensity Erbium:YAG (solid-state erbium-doped yttrium aluminum garnet) lasers emitting light at 2940 nm are routinely used for skin ablation. Each pulse can ablate a reproducible amount of tissue; thus, the pore depth can be controlled [39,46]. Using this technology, Pantec Biosolutions AG (Ruggell, Liechtenstein) developed the P.L.E.A.S.E^®^ system (Precise Laser Epidermal System) for delivery of low and high molecular weight molecules. The device can create an array of 150 μm diameter micropores on a small skin area. The pore depth is controlled by modulating (i) the number of pulses per pore and (ii) pulse energy or fluence (J/cm^2^). The latter depends on the pulse duration (μs) and repetition rate (Hz). To obtain minimally invasive painless ablation, the pore depth must be limited so as not to reach the sensitive nerve endings situated in the dermis. Finally, the number of pores created per unit area determines the fraction of skin surface that is removed, and this is defined as the fractional ablated area (%). Thus, selectively ablating superficial layers of skin would hypothetically provide direct access to epidermal cells able to express the target protein.

For each of the different treatment conditions, upon completion of the formulation application period, the delivery site was harvested using a 10 mm diameter punch (i.e., skin surface area was 3.14 cm^2^), and biopsies were placed in a culture medium. All samples were cultured in an incubator (37 °C; 5% CO_2_; 100% RH) for 24 h or 48 h.

### 2.4. Sample Processing

#### 2.4.1. Snap Freezing and Cryosectioning

Harvested skin biopsies were snap-frozen using an optimized technique modified from Lapteva et al. [47,48]. Briefly, the skin biopsy was positioned to flatten the stratum corneum against the flat part of a plastic mold, which was placed on frozen ethanol cooled by liquid nitrogen and the periphery of the biopsy was gently pressured with a pipette tip to keep the skin flat and avoid sample retraction. OCT compound was poured in the mold once the sample stuck to it. Upon complete solidification of OCT compound, the samples were unmolded and stored at −80 °C until further processing. Cryopreserved biopsies were mounted in a cryotome (Microm HM 560 Cryostat, Walldorf, Germany) to obtain horizontal (XY plane) 10-μm sections. A total of 40 sections was taken from each sample, theoretically, going from the skin surface to a depth of 400 μm, providing tissue samples from the stratum corneum, viable epidermis, and upper dermis. Each section was collected on a Superfrost™ glass slide (Thermofisher Scientific, Plan-les-Ouates Switzerland).

#### 2.4.2. Tissue Characterization: Hematoxylin-Eosin Staining

Skin sections of an untreated skin sample were stained with standard H&E protocol to characterize each section histology (20-µm thickness). Sections were visualized with a stereomicroscope (LEICA S6D, Leica, Heerbrugg, Switzerland).

#### 2.4.3. Cy5 Labeled mRNA Delivery: Staining

The following staining protocol was used for samples containing Cy5 labeled mRNA. Skin sections were fixed in 4% formaldehyde solution and washed once in PBS. They were subsequently stained with 300 nM DAPI (4′,6-diamidino-2-phénylindole, Axonlab, Le Mont-sur-Lausanne, Switzerland) solution for 5 min and underwent a final wash step in PBS before mounting in Eukitt mounting medium.

#### 2.4.4. eGFP Expression: Immunochemical Staining

This staining protocol was used for samples after delivery of eGFP mRNA to localize the expression of eGFP. 

The sections were fixed for 5 min in methanol:acetone (1:1) at −20 °C and subsequently washed three times in PBST at RT. All of the following steps were performed at RT. Sample slides were exposed to 3% hydrogen peroxide in PBS for 10 min. They were subsequently washed twice in PBST and once in 0.5 M Tris buffer (pH 8). Unspecific binding was prevented by blocking the section with 4% normal goat serum (NGS) in PBS for 40–60 min. Then, the primary antibody (mouse anti-GFP, clone 9F9.F9, mIgG1; 1–3 µg/mL) in 4% NGS was applied to the sections for 40–60 min. Slides were washed three times in PBST. The secondary antibody (Alexa Fluor™ 594 goat anti-mouse IgG [H + L]; 5 µg/mL) in 4% NGS was applied to the sections and incubated for 60 min. Another wash step (3 × 5 min) in PBST was performed. Finally, nuclei were stained with DAPI as mentioned above. Slides underwent a final wash in PBS (two times, 5 min) and once in 0.5 M Tris buffer (pH 8). Samples were mounted in Mowiol^®^ solution (Mowiol: Glycerol: water: 0.2 M Tris buffer; 9:23:23:45) and left to dry overnight in dark.

#### 2.4.5. Widefield Microscopy and Confocal Laser Scanning Microscopy

Image acquisition was performed in the Bioimaging Core Facility, Faculty of Medicine, University of Geneva. To obtain widefield images, skin sections were examined with a Zeiss Axioscan.Z1 Microscope (Carl Zeiss; Jena, Germany). Excitation and emission wavelengths for (i) DAPI were 365 and 445 nm and (ii) Cy5 and secondary antibody AF594 were 640 and 690 nm, respectively. Images were acquired using a Hamamatsu Orca Flash 4 monochrome fluorescence camera (intensity 70%) (Hamamatsu Photonics, Solothurn, Switzerland). Exposure times were 20 ms for DAPI and 70 ms for Cy5 and AF594. The 20× objective was used (Plan-Apochromat 20×/0.8).

To obtain confocal planes and study the co-localization of signals, skin sections were examined with a Zeiss LSM700 confocal microscope (Carl Zeiss, Jena, Germany). Excitation and emission wavelengths for (i) DAPI were 405 and 465 nm and (ii) Cy5 and secondary antibody AF594 555 and 690 nm, respectively, using a 63× oil objective (Plan-Apochromat 63×/1.4 Oil). Although AF594 emitted in the red region of the spectrum, the signal is shown as green in the present work for more clarity (since it was used to monitor eGFP).

#### 2.4.6. Data Processing

Images were acquired using Zen 2.3 pro software (Carl Zeiss AG, Feldbach, Switzerland) and processed using Image J 1.52n software (Rasband, W.S., ImageJ, U. S. National Institutes of Health, Bethesda, MA, USA). Semi-quantitative data as a function of skin depth was extracted from processed images using the pixel count function in Image J 1.52n software. 

## 3. Results and Discussion

### 3.1. Tissue Characterization

To enable a three-dimensional visualization of mRNA delivery and protein expression, ex vivo porcine skin samples were cryotomed according to the previously described cutaneous biodistribution method (CBM) [47], to produce 40 lamellae, each with a thickness of 10 µm going from the skin surface to the lower dermis (400 µm) (Figure 1a). Despite the initial sections being somewhat brittle due to skin surface rugosity and lack of tissue cohesion, widefield FM allowed the acquisition of high-resolution images of each complete lamella (i.e., full-size), enabling a clear visualization of the distribution of the Cy5 labeled mRNA and eGFP expression in each skin layer. “Stacking” of the results from each layer enabled a depth profile to be obtained, illustrating the spatial distribution in the stratum corneum, viable epidermis and dermis down to a depth of 400 µm (Figure 1b).

DAPI staining was used to indicate indirectly the histological region of skin: the arrangement of cell nuclei and their density allowed the identification of the heterogeneous skin layers. H&E staining was used to confirm the different histological skin structures (Figure 1c–e). Figure 1c,d show typical histological patterns found in horizontal sections of skin. FM observations indicate that stratum corneum appears as lightly stained threads (Figure 1c), while viable epidermis can be easily identified by the dense packing of cell nuclei of keratinocytes in DAPI stained samples and hematoxylin-rich regions in the H&E staining. Skin is a heterogeneous organ where every layer is invaginated into each other: epithelial cells co-exist with dermal tissue in several slices, separated by the dermo-epithelial junction (Figure 1c,d). This papillary dermis is characterized by the sparser distribution of the cell nuclei from fibroblasts. In the deeper reticular dermis, epithelial cells can be observed in the form of the lining of pilosebaceous units (PSU) and sweat ducts (Figure 1e). The reticular dermis also presents groups of hematoxylin stained cells corresponding to the vascular tissue [49]. In skin, blood vessels are often accompanied by the lymphatic network [50] (Figure 1e). In subsequent observations, only DAPI staining was used to identify histological skin structures. 

### 3.2. Effect of Different Delivery Techniques on Skin Distribution of Cy5 Labeled mRNA and eGFP Expression

Images obtained by the combination of all DAPI stained sections after the delivery of Cy5 labeled mRNA and eGFP expression using different formulations and delivery techniques were compared. All montage images are composed of 5 rows and 8 columns starting with section 1 (“nominal” depth going from skin surface to10 µm) in the upper left and finishing by section 40 (“nominal” depth going from 390 to 400 µm) in the lower right. The scale is not indicated as each section has a diameter of 10 mm. All missing or compromised sections have been replaced by the black background to keep the section numbering order consistent. Cy5 labeled mRNA appears in red (left panels), while the eGFP signal appears in green (right panels). Control samples (that underwent no mRNA administration) did not show any Cy5 red (Figure 2) and a faint green background green signal. Thus, the observation method was considered as specific and accurate for both compounds.

#### 3.2.1. Intradermal Injections

A typical Cy5 signal at the injection site, i.e., in the dermal layers, could be observed 24 h after intradermal injection of Cy5 labeled mRNA (1 µg dose) using a polymeric transfection agent (Figure 3a). It spread from section 17 (170 µm) down to the last section (400 µm) with increasing intensity, putatively going even deeper in the dermis. A star-shaped pattern could be evidenced in the last sections corresponding to the mRNA diffusion from the injection site (Figure 3a). The Cy5 signal in deeper dermal layers seems to fade after 48 h of skin culture, pointing to its possible degradation or diffusion from the injection site (Figure 3c). No Cy5 signal was evidenced at the section’s periphery (more than 2 mm away from the injection site). The localization of Cy5-mRNA was both extracellular and superimposed with the cell nuclei (i.e., indeed internalized into the cell) (Figure 3i). The colocalization was confirmed by confocal microscopy in three dimensions (Appendix A).

Interestingly, eGFP expression did not follow the same pattern as mRNA distribution (Figure 3b): eGFP expression was observed to a greater extent in the epidermis than in the dermis. It further increased in the epidermis 48 h after the mRNA injection (Figure 3d). Figure 3e–h present a magnification of sections 8 (80 µm—epidermis) and 38–39 (380–390 µm reticular dermis), respectively. Green signal originated mainly from the epidermal keratinocytes (Figure 3f), from appendageal structures, e.g., PSU and sweat glands, vascular structures, and, to a lesser extent, from dermal fibroblasts (Figure 3h). In contrast, the Cy5 mRNA signal was focused at the injection site and was only visible starting at a depth of 120 μm below the skin surface, and eGFP expression occurred in every layer of skin: to a great extent in the epidermis, including the sample periphery. This lack of co-localization of the mRNA detection site and actual protein expression was unexpected. It seems clear that eGFP expression is performed preferentially by keratinocytes and epithelial cells of skin appendages. 

However, the mechanisms by which mRNA—delivered to the deep dermis—reached the superficial layers of skin and was internalized by keratinocytes remain uncertain. Several hypotheses can be formulated: (i) mRNA reached the epidermis via the reflux induced by high back-pressure after injection, (ii) mRNA migrated passively back to the epidermis via appendageal structures encountered in deeper dermis, or (iii) mRNA migrated passively back to the epidermis via lymphatic/blood vessel network present in the dermis. Given the presence of eGFP at the sample periphery, there could be a preference for the last hypothesis because of the extensive presence of vascular/lymphatic network in the dermis.

The 600 µm MicronJet^™^600 hollow microneedle from (NanoPass Technologies Ltd.; Nes Ziona, Israel) is an FDA- and EMA-approved array of three silicon 600 μm microneedles attached to a tip that can be easily mounted on a syringe. They are designed to facilitate accurate intradermal injection. Figure 4 shows the mRNA delivery yielded by MicronJet™600. The three injection spots can be clearly distinguished in Cy5 labeled mRNA (Figure 4a) and eGFP (Figure 4b) signals. However, and consistent with traditional ID injection, the same phenomenon is observed: eGFP seems to be expressed away from the injection site by epidermal keratinocytes (skin surface to 120–150 μm depth Figure 4b), by epithelial cells in appendageal structures, PSU and sweat glands, vascular cells, and, to a lesser extent, by dermal fibroblasts (Figure 4d). 

#### 3.2.2. Jet Injection

A needleless injection system, Dermojet^®^ (AKRA DERMOJET; Pau, France) was also used to deliver the mRNA formulations. Liquid-jet injectors use compressed gas or a spring to generate a high-velocity jet (with velocities ranging from 100 to 200 m/s) propelled from a nozzle with a pressure of 9,810,000 N/m^2^ (i.e., about 1420 psi or 98 bar). Depending on the jet velocity and orifice diameter, the jet can be delivered into the intradermal region or deeper [39]. The mRNA was delivered using two different heads: one with a single nozzle and one with a triple nozzle.

Figure 5a–d shows the mRNA delivery and eGFP expression yielded by this delivery method. The Cy5 signal is only faintly visible in the last (i.e., deepest) skin sections. Figure 5 presents details of sections 32 and 36 for Dermojet^®^ 1-head jet injection and Dermojet^®^ 3-head jet injection, respectively. It appears that the use of the triple-headed jet injector results in a broader delivery of the mRNA, allowing it to reach the sample periphery. However, these different delivery modalities result in no change in the expression pattern of the eGFP protein: as for the intradermal injections, it seems to be expressed away from the injection site by epidermal keratinocytes, by epithelial cells in appendageal structures as PSU, and sweat glands, vascular tissues, and weakly dermal fibroblasts (Figure 5).

#### 3.2.3. Fractional Laser Ablation

The fractional laser ablation using an Er:YAG laser (erbium-doped yttrium-aluminum-garnet laser) emitting a wavelength of 2940 nm [51] enables the precise removal of the superficial layers of the epidermis by thermal excitation and evaporation of the water molecules present, thereby creating micropores for minimally invasive drug delivery [5,52,53]. The wavelength corresponds to an absorption band of water in the infrared spectrum. Upon exposure, water molecules vibrate and heat-up. Four effects may occur: heating (37–60 °C), coagulation (60–65 °C), drying (90–100 °C), and vaporization (>100 °C). Using this technology, the P.L.E.A.S.E.^®^ system (Precise Laser Epidermal System) was developed by Pantec Biosolutions AG (Ruggell, Liechtenstein) for delivery of low and high molecular weight therapeutics. The device can create an array of 150 µm diameter micropores on a small skin area. The pore depth is monitored by modulating (i) the number of pulses per pore and (ii) pulse energy or fluence (J/cm^2^). The latter depends on the pulse duration (µs) and repetition rate (Hz). To obtain minimally-invasive painless ablation, the pore depth must be limited so as not to reach the sensitive nerve endings situated in the dermis. Finally, the number of pores created per unit area determines the fraction of skin that is ablated, and this is defined as the fractional ablated area (%) [44,54].

The skin was porated using 35.4 J/cm^2^ (1.2 W 175 µs; 200 Hz, 3 pulses per pore, fractional ablated area of 7 % on a disc of 10 mm diameter) before topical application of mRNA containing formulations. It has already been demonstrated that fractional laser ablation using the P.L.E.A.S.E device could significantly increase the skin deposition and permeation of an anti-CD29 monoclonal antibody in vitro when the laser fluence was above 35.1 J/cm^2^ [44]. Figure 6 shows the mRNA delivery and eGFP expression yielded by this delivery method. Pores could be visualized up to a depth of 110 μm; thus they, were not reaching the dermis with its sensitive nerve endings. Cy5 signal was observed at the skin surface and inside the pores only. No deeper presence of Cy5 labeled mRNA was evidenced. Indeed, it seemed that mRNA was barely able to diffuse away from the pores (Figure 6c–e). This was confirmed by the fact that, in contrast with intradermal delivery techniques, eGFP was poorly expressed by fractional laser ablated samples despite a higher dose: expression occurred in the epidermis only in the neighborhood of the pores (Figure 6f) and faintly in the dermal vascular tissues. Interestingly, no expression in appendageal structures was observed after fractional laser-assisted delivery (Figure 6g). 

Table 2 summarizes the delivery sites and type of cells expressing the protein of interest after delivery using different modalities.

The magnified microscopic observations of these cells are presented in Appendix A. When polymeric and liposomal transfection agents were compared with respect to ID injections, jet injector, and fractional laser delivery systems, both transfection agents were found to be equally effective.

### 3.3. Semi-Quantification of Cy5 Labeled mRNA Delivery and Subsequent eGFP Expression

To better compare the different delivery techniques and to provide more quantitative data about the mRNA delivery and eGFP expression, microscopic pictures were processed to enable specific pixel counting of Cy5 signal (red) and GPF signal (green) in every skin section. Figure 7 shows pixel counts as a function of skin depth. This semi-quantitative method appeared to be valid since the untreated skin resulted in low pixel count (due to background noise) for both Cy5 and eGFP signals (Figure 7a,b). In agreement with visual evaluation, 24 h after the intradermal injection, Cy5 labeled mRNA signal increased with skin depth, reaching a maximum in the last section. 48 h post-injection this signal started to decrease in the dermal layers (below 160 µm; Figure 7a,b). The corresponding eGFP expression estimated by pixel counting points to a high expression in epidermal layers at 24 h and a maximal expression at 48 h between 40 and 120 µm of skin depth (viable epidermis). This again confirms the visual observations made previously. The microneedle-mediated injection yielded similar delivery and expression profiles to the intradermal injection (Figure 7c,d). Both jet injector delivery techniques yielded similar mRNA delivery and eGFP expression (Figure 7e,f). Fractional laser ablation delivered Cy5 labeled mRNA to superficial layers of epidermis mainly; however, eGFP expression was decreased in comparison with the ID injection (Figure 7g,h). The use of liposomal transfection agent did not produce any difference in Cy5 labeled mRNA delivery nor eGFP expression, regardless of the delivery technique used, apart from a slightly deeper signal after ID injection, probably due to a different injection depth (Figure 7i–l). 

The semi-quantitative assessment of mRNA distribution and eGFP expression as a function of depth are highly consistent with the visual observations from the microscopy studies. This representation allows comparison of different formulations in terms of depth profiles and areas under the curve. Figure 8 summarizes the total delivery and expression resulting from the different techniques. The best eGFP expression was undeniably yielded by the ID injection. 

### 3.4. Clinical/Preclinical Implications

Several recent studies have been performed to evaluate the efficacy of mRNA intradermal injections for the in situ expression of therapeutic proteins [33,38,55]. Golombek et al. studied the ID delivery of a 900 bases hGLuc-encoding—Cy3 labeled mRNA and subsequent expression of humanized Gaussia luciferase using magnetic microspheres or lipoplexes as transfection enhancers and the MicronJet™600 microneedle system as an injection device [38]. The depths of the injection sites were determined by injecting metallic microspheres using the MicronJet™600 microneedle system and their distribution was studied. Microspheres were localized in the dermis at approximate depths ranging from 200 to 1300 µm with the main bolus being at a depth of 800 µm. These depths are consistent with what can be observed with Cy5-mRNA delivery using MicronJet™600 injection and single intradermal injection (Figure 3 and Figure 4). Golombek et al. performed the detection of the hGLuc-encoding—Cy3 labeled mRNA after intradermal injection: unsurprisingly, it was detected in the dermis; however, the visualization was done using a standard cross-sectioning, and no visualization of subsequent protein expression was performed [38]. Shortly after, Pehrsson et al. investigated the ID delivery of a modified VEGF-A165 mRNA to express human VEGF-A protein locally in vivo in rabbit and pig to treat vascular diseases. It is worth noting that no transfection agents were used; however, the injection device used for the intradermal injection was not described. Human VEGF-A protein expression was assessed using microdialysis and quantified in injection site biopsies. The expression was successfully detected in microdialysis eluates and skin tissues; however, neither mRNA distribution in skin nor that of the protein were investigated [55]. This study was immediately complemented by a first in man trial: Gan et al. demonstrated that, after the intradermal injection of naked mRNA encoding VEGF-A165 in patients (n = 27 in treated group) with Type 2 diabetes, elevated VEGFA protein levels could be detected at mRNA-treated sites [33]. Clinically relevant enhancement in basal skin blood flows up to 7 days after administration could be detected using laser Doppler fluoximetry and imaging.

In the present study, the use of a broad panel of delivery techniques, the visualization of Cy5 labeled mRNA delivery, and eGFP expression yielded much information about the fate of mRNA in the skin. According to the different delivery/expression patterns that were observed, delivery techniques could be divided into two groups: dermal targeting techniques (i.e., intradermal injections, MicronJet™600 hollow microneedle injections and Dermojet^®^ jet injectors) and the techniques targeting the epidermis (i.e., fractional laser ablation). Considering the dermal techniques—it appears that, once in the dermis, mRNA can be internalized and expressed by fibroblasts, appendageal epithelium, and vascular/lymphatic tissue. The high epidermal expression of eGFP suggests that non-internalized mRNA probably retro-migrates via the vascular/lymphatic tissue into the epidermis and is expressed there. Epidermal delivery using fractional laser ablation yielded unexpected results—it appears that mRNA could not migrate efficiently away from the pores. The specific fractional laser ablation conditions (35.4 J/cm^2^ (1.2 W 175 µs; 200Hz, 3 pulses per pore)) allowed the successful delivery of an antibody to the skin [44]. However, biochemically an antibody and mRNA are very different, especially in terms of charge: nucleic acids carry many negative charges, and it is possible that an electrostatic interaction could retain the mRNA at the pore surface, rendering it less available for further diffusion into the skin. It is also possible that the pore suffers from tissue coagulation, thus creating a barrier to mRNA diffusion [44,56,57]. Given that the pore depth appeared to be of 110 µm, it might be that mRNA penetration into skin was hindered by the remaining tight junctions present in the lower epidermis and lamina basale [58]. Therefore, the modalities of fractional laser-mediated epidermal delivery need to be explored further, together with the elucidation of the role of tight junctions [59].

In the present study, we found that intradermal injection yielded the highest mRNA distribution and greatest protein expression in the skin, followed by MicronJet™600 hollow microneedle injection, as compared to other delivery techniques. In both cases, the entire mRNA formulation was injected directly in the dermal region where a depot was created, and, subsequently, mRNA was distributed throughout the tissue. This might have been a key factor in the higher mRNA distribution and greater protein expression observed; in the case of the jet injector, the complete volume of mRNA formulation could not be delivered to the skin as some part of the formulation bounced back off the skin surface. In the case of fractional laser ablation, the mRNA formulation was applied topically on the porated area of the skin; therefore, the mRNA had to diffuse from the formulation to the pores and finally distribute itself in the tissue. To avoid visual bias during analysis of the images, as every person perceives images differently, the semi-quantitative methodology was found to be very important in that it enabled robust and uniform conclusions to be drawn. This provides a simple and effective approach to study mRNA delivery efficacy, both in terms of actual mRNA fate in the tissue, as well as the extent of protein expression. In real-world applications, ID injection and the MicronJet™600 microneedle system would be relatively suitable delivery techniques being easily available, providing a reproducible/predictable result and with no need of sophisticated devices which could be challenging when larger populations need to be addressed, for instance, in the case of mass vaccinations in a pandemic.

## 4. Conclusions

In this study, we observed that different delivery techniques were able to target different regions in the skin. Dermal delivery techniques, such as ID injection, the MicronJet™600 microneedle system, and jet injectors, can be used to have targeted mRNA expression in keratinocytes, appendageal epithelium, and vascular endothelium, whereas the fractional laser ablation could be used for targeted expression in vascular endothelium and, to some extent, in keratinocytes. The cutaneous biodistribution method provides a view of the three-dimensional distribution of both the ribonucleic acid and subsequent protein expression following mRNA delivery using several formulations and delivery techniques. In that sense, the present study is unprecedented as it allowed us to visualize the correlation between the site of delivery of the mRNA and subsequent expression. Moreover, a semi-quantitative method has been developed to compare the different delivery modalities in terms of delivery and expression as a function of skin depth, which enabled robust unbiased conclusions to be drawn. Confirmative studies in human skin ex vivo and in pigs in vivo are planned.

## Figures and Tables

**Figure 1 pharmaceutics-14-00151-f001:**
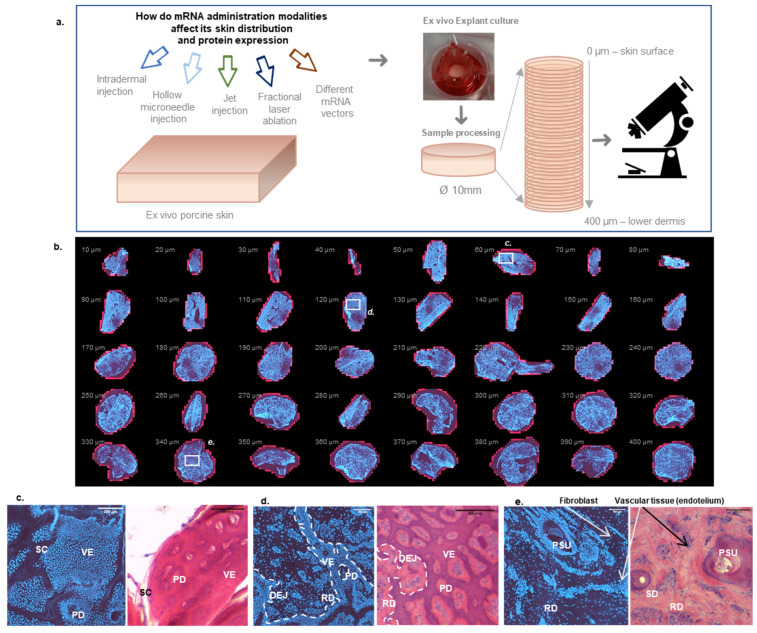
Histological characterization of skin horizontal sections. (**a**) Experimental Workflow: each sample that underwent mRNA ex vivo delivery was horizontally cryotomed into 40 slices of 10 µm thickness prior to microscopic observation. (**b**) Montage of all 40 DAPI-stained skin sections after FM observation (Ø10 mm; untreated skin): Cy5 background signal (red) and DAPI signal (blue), each sample in the montage has a 10 mm diameter. Magnified slices when stained with DAPI and with H&E: (**c**) Comparative histology of slice 6; 60 µm depth—viable epidermis. (**d**) Comparative histology of slice 12; 120 µm depth—at the dermo-epidermal junction. (**e**) Comparative histology slice 34; 340 µm depth—deep reticular dermis. Scale bar (**c**–**e**): 200 µm. Annotations: SC: Stratum corneum, VE: viable epidermis PD: papillary dermis DEJ: Dermo-epithelial junction RD: Reticular dermis PSU: pilosebaceous unit SD: sweat duct. Individual Fibroblasts and vascular endothelial tissue could also be observed.

**Figure 2 pharmaceutics-14-00151-f002:**
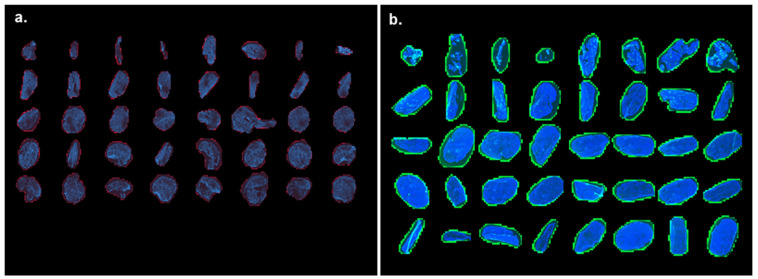
Background signals. (**a**) Cutaneous distribution of background Cy5 and (**b**) eGFP signals in untreated skin samples, i.e., in the absence of mRNA application. The images show background Cy5 (red), background eGFP (green) and DAPI (blue) signal at 24-h skin culture. (High resolution images of the above and all subsequent images) are available in the Appendix A.

**Figure 3 pharmaceutics-14-00151-f003:**
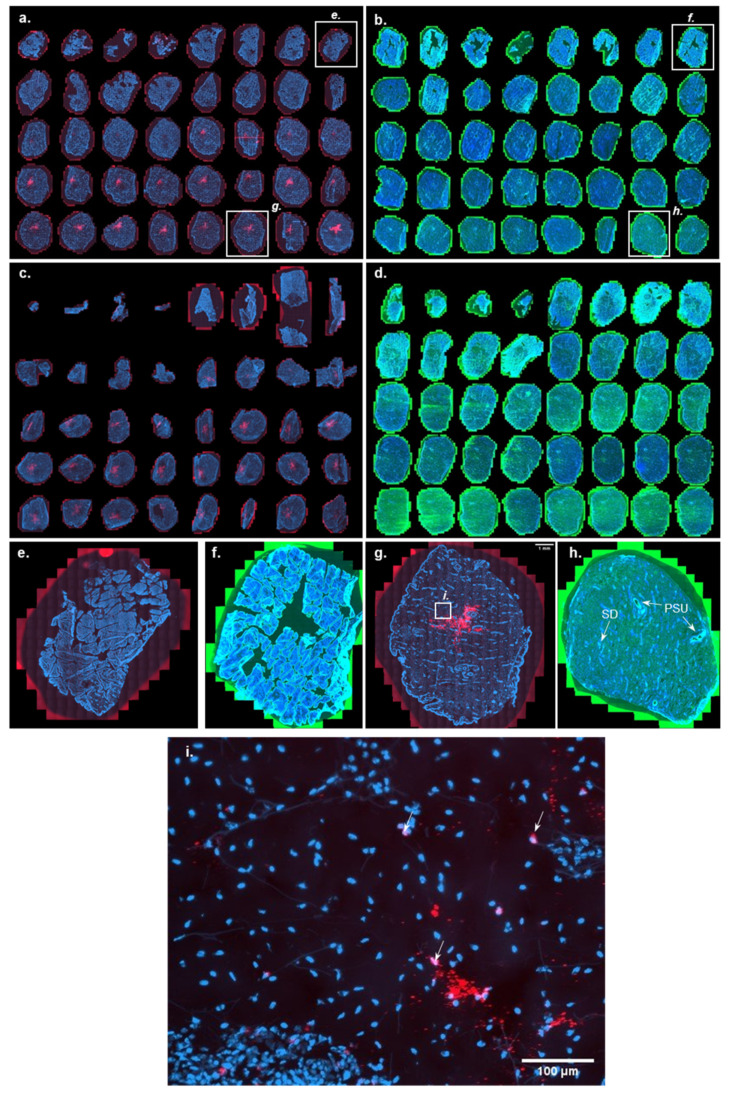
Cutaneous distribution of Cy5 labeled mRNA and eGFP after ID delivery of the former and eGFP expressing mRNA using conventional ID injection. The images show the Cy5 labeled mRNA (red), eGFP (green), and DAPI (blue) signal 24 h and 48 h after ID injections of the polymer formulations containing 1 µg of either the Cy5 labeled mRNA or the eGFP expressing mRNA using a syringe equipped with a 26 G × 3/8″ needle. (**a**) Cy5 labeled mRNA signal going from a skin depth of 10 to 400 µm, 24 h after ID injection. Cy5 signal is seen at depth ≥ 200 µm, i.e., in the dermis (**b**) eGFP signal going from a skin depth of 10 to 400 µm, 24 h after ID injection of eGFP expressing mRNA. eGFP signal can be seen in the epidermis, suggesting that diffusion of mRNA occurs after injection into the dermis to the sites of protein expression. (**c**) Cy5 labeled mRNA signal going from a skin depth of 10 to 400 µm, 48 h after ID injection. (**d**) eGFP signal going from a skin depth of 10 to 400 µm, 48 h after ID injection. eGFP signal appears to be appreciably more intense at the longer time point. (**e**,**f**) are close-up images of the Cy5 labeled mRNA and eGFP signals from the epidermis (section 8, corresponding to a depth of 80 µm) after 24 h. (**g**,**h**) are close-up images of the Cy5 labeled mRNA and eGFP signals from the dermis (section 38, corresponding to a depth of 380 µm) after 24 h. PSU: pilosebaceous unit SD: sweat gland/duct. (**i**) Magnification of section 38 (corresponding to a depth of 380 µm) after 24 h showing non-internalized mRNA and mRNA internalized in fibroblasts (white arrows). For panels (**a**–**d**), high resolution images are available in Appendix A.

**Figure 4 pharmaceutics-14-00151-f004:**
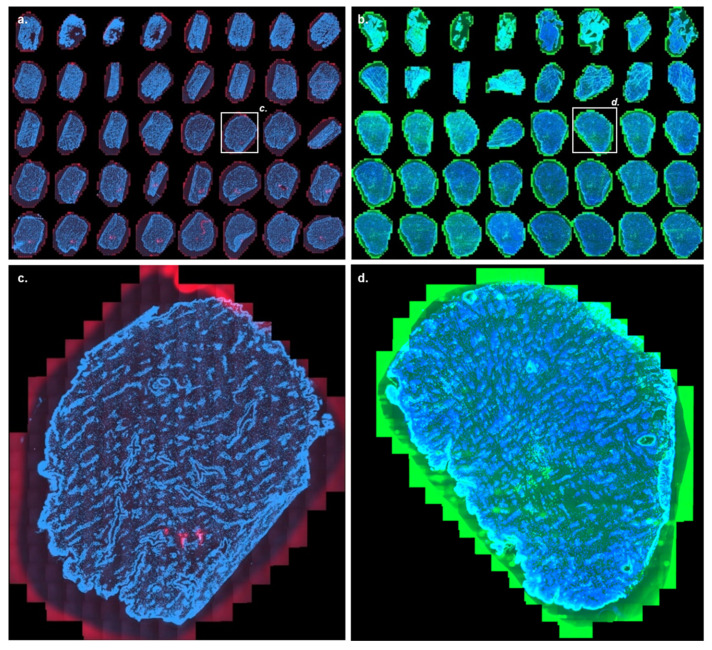
Cutaneous distribution as a function of depth of Cy5 labeled mRNA and eGFP after ID delivery of the former and eGFP expressing mRNA using the MicronJet™600 hollow microneedle injector. The images show the Cy5 labeled mRNA (red), eGFP (green), and DAPI (blue) signal 48 h after injection of the polymer formulations containing 1 µg of either Cy5 labeled mRNA or eGFP expressing mRNA using the hollow microneedle MicronJet™600 injector. (**a**) Cy5 labeled mRNA signal going from a skin depth of 10 to 400 µm. As with conventional ID injection, the Cy5 signal only begins to be visible in the dermis. (**b**) eGFP signal going from a skin depth of 10 to 400 µm after injection of eGFP expressing mRNA; again, as for the conventional ID injection, the eGFP signal is present in the viable epidermis, indicating that the site of mRNA delivery is not necessarily the site of protein expression. (**c**,**d**) Close-up images of the Cy5 labeled mRNA and eGFP signal from the dermal region (section 22, corresponding to a depth of 220 µm), 24 h after MicronJet™600 injection of the Cy5 labeled mRNA and eGFP expressing mRNA. For panels (**a**,**b**), high resolution images are available in Appendix A.

**Figure 5 pharmaceutics-14-00151-f005:**
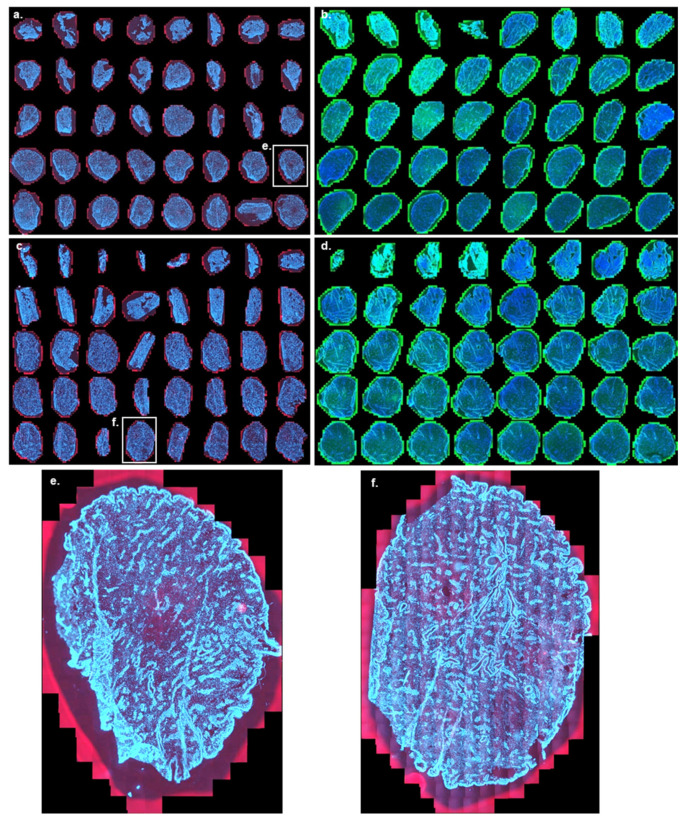
Cutaneous distribution as a function of depth of Cy5 labeled mRNA and eGFP after delivery of the former and eGFP expressing mRNA using the Dermojet^®^ jet injector system equipped with either 1 or 3 heads. The images show the Cy5 labeled mRNA (red), eGFP (green), and DAPI (blue) signal 24 h after delivery of the polymer formulations containing 1 µg of either Cy5 labeled mRNA or eGFP expressing mRNA using the Dermojet^®^ system. (**a**,**b**) Images of the Cy5 labeled mRNA and eGFP signals going from a skin depth of 10 to 400 µm after delivery of either the Cy5 labeled mRNA or the eGFP expressing mRNA using the Dermojet^®^ 1-head device; (**c**,**d**) Images of the Cy5 labeled mRNA and eGFP signals going from a skin depth of 10 to 400 µm after delivery of either the Cy5 labeled mRNA or the eGFP expressing mRNA using the Dermojet^®^ 3-head device. The results with the Dermojet^®^ jet injector are consistent with those seen after ID injection with Cy5 mRNA signal being predominantly present in the dermis and the eGFP signal being also strongly seen in the epidermis. Thus, the jet injector, despite being applied to intact skin, deposits mRNA in the dermis. (**e**,**f**) Close-up image of the Cy5 labeled mRNA signal from the dermal region—section 32, corresponding to a depth of 320 µm—after application of the Dermojet^®^ 1-head device and—section 36, corresponding to a depth of 360 µm—after using the Dermojet^®^ 3-head jet injector. The effect of using the 3-head device is clearly visible with the three discrete zones of Cy5 labeled mRNA. For panels (**a**–**d**), high resolution images are available in Appendix A.

**Figure 6 pharmaceutics-14-00151-f006:**
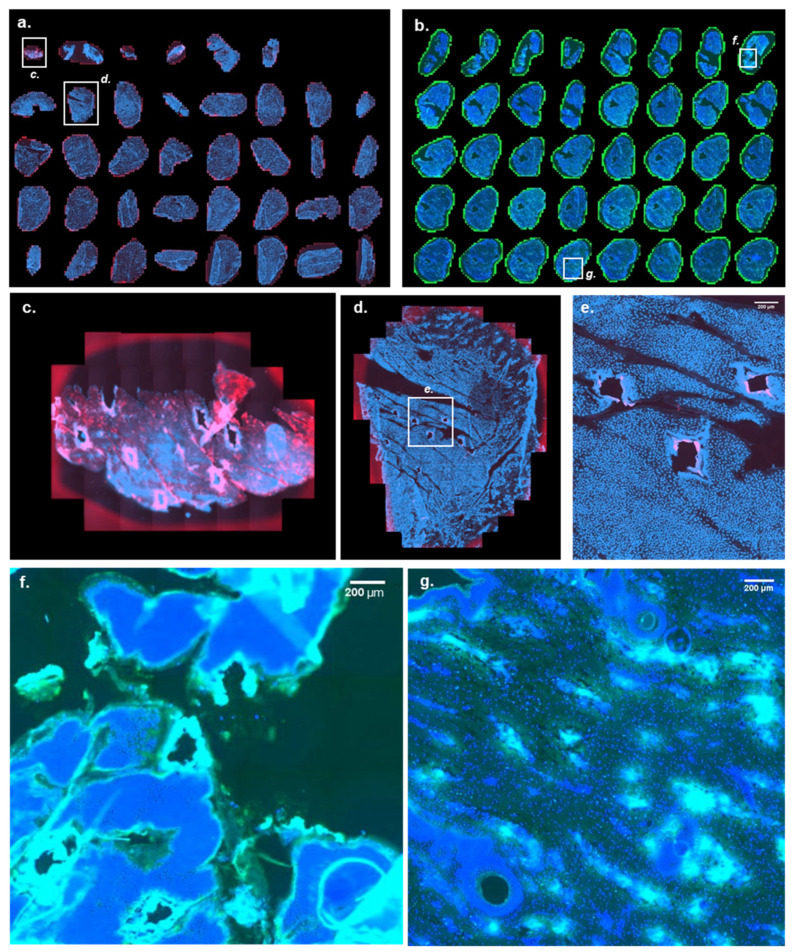
Cutaneous distribution as a function of depth of Cy5 labeled mRNA and eGFP after delivery of the former and eGFP expressing mRNA upon application to the skin surface after Er:YAG fractional laser ablation. The images show the Cy5 labeled mRNA (red), eGFP (green), and DAPI (blue) signal 24 h after delivery of the polymer formulations containing 1 µg of either Cy5 labeled mRNA or eGFP expressing mRNA into skin after fractional laser ablation. (**a**,**b**) are images of the Cy5 labeled mRNA and eGFP signals going from a skin depth of 10 to 400 µm after delivery of either the Cy5 labeled mRNA or the eGFP expressing mRNA. Laser-assisted drug delivery is a two-step process with the formulation being applied to the skin surface after fractional laser ablation. Unlike the other technologies, mRNA delivery and deposition are not targeted to the dermis. Hence, the presence of Cy5 labeled mRNA and eGFP in the upper epidermis is to be expected. (**c**) Close-up image of the Cy5 labeled mRNA signal from the uppermost epidermal region—section 1, corresponding to a depth of 10 μm. (**d**) Closer view of the lower epidermal region—section 10, corresponding to a depth of 10Im. (**e**) Increased magnification of the micropores created by fractional laser ablation present in section 10, at a depth of 100 μm. The Cy5 labeled mRNA signal is localized around the micropores. (**f**) eGFP expression around the micropores in the epidermis (section 8, corresponding to a depth of 80 μm) and (**g**) eGFP expression in the dermis (section 36, corresponding to a depth of 360 μm) confirms that the eGFP expressing mRNA was able to diffuse away from the micropores to the sites of protein expression in the dermis. For panels (**a**,**b**), high resolution images are available in Appendix A.

**Figure 7 pharmaceutics-14-00151-f007:**
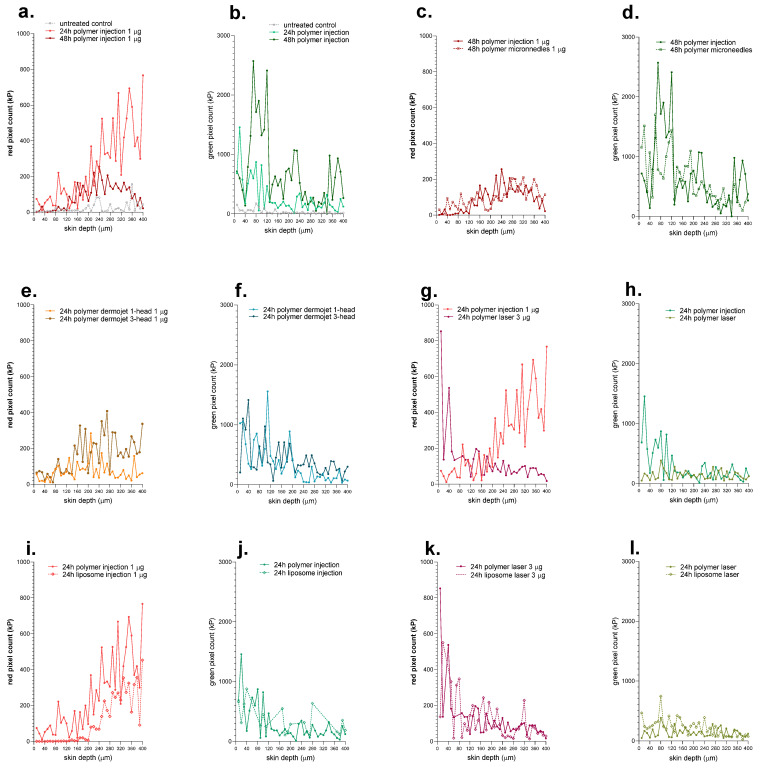
Semi-quantitative analysis of the delivery of Cy5-labeled mRNA and eGFP expression following delivery of eGFP expressing mRNA using the different delivery technologies. The pixel count as a function of depth was used to provide a measure of the delivery of Cy5-labeled mRNA (**a**,**c**,**e**,**g**,**i**,**k**) and the extent of eGFP expression (**b**,**d**,**f**,**h**,**j**,**l**) achieved using the different delivery technologies and experimental conditions. (**a**,**b**) temporal variation of Cy5 labeled mRNA and eGFP expression after ID injection shows opposite trends—loss of Cy5 labeled mRNA with time, especially in the dermis, but a significant increase in eGFP expression. (**c**,**d**) Confirmation that intradermal delivery with the MicronJet™600 hollow microneedle injector is similar to an intradermal injection. (**e**,**f**) The use of the Dermojet^®^ 3-head system appeared to deliver more Cy5 labeled mRNA to the dermis, but eGFP expression was similar. (**g**,**h**) show the striking difference in delivery of Cy5 labeled mRNA and eGFP expression following administration by ID injection and after fractional laser ablation; Cy5 labeled mRNA is present principally in the dermis after the former and in the superficial epidermis with the Iatter. (**i**,**j**) comparison of polymeric and liposomal transfection agents on the delivery of Cy5 labeled mRNA using ID injection. Although the trends are similar with increasing signal found in the dermis, delivery of Cy5 labeled mRNA using the polymeric formulation appears to be better and is significantly greater in the epidermis and upper dermis. In contrast, despite mRNA delivery targeting the dermis, the eGFP signal was far more intense in the epidermis, suggesting that mRNA was diffusing post-delivery as this was the principal site of protein expression. (**k**,**l**) comparison of polymeric and liposomal transfection agents on delivery of Cy5 labeled mRNA using fractional laser ablation; trends were similar with Cy5 labeled mRNA predominantly found in the superficial epidermis and decreasing with depth, whereas eGFP expression was low and more constant throughout (images shown in Figure 5 suggest that it occurred in proximity to the micropores but that it was possible for mRNA to also be expressed in the dermis).

**Figure 8 pharmaceutics-14-00151-f008:**
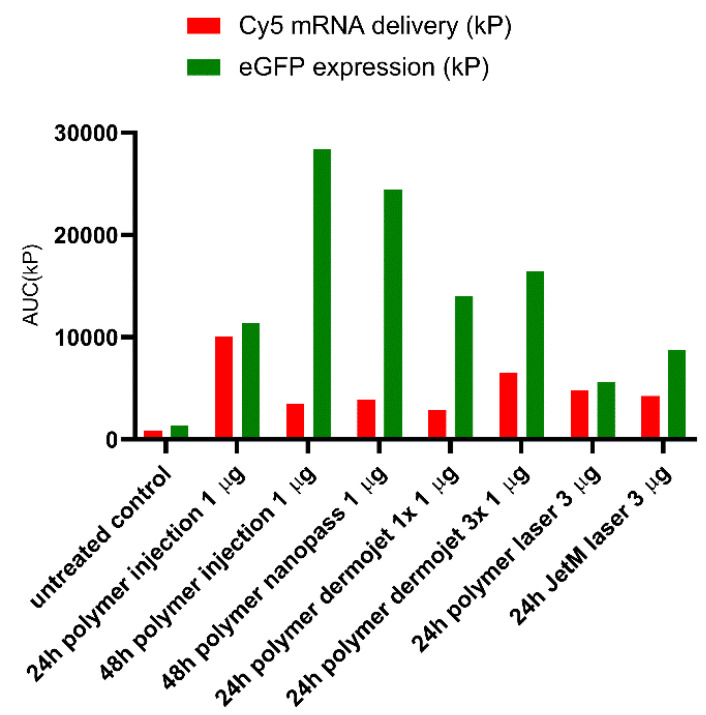
Delivery of Cy5 labeled mRNA and expression of eGFP as assessed by the AUC of pixel count-depth profiles. Cy5 labeled mRNA (red) and eGFP (green) area under the curve (AUC; based on the total pixel count vs depth profile) for the different delivery modalities. Striking and contrasting temporal effects were evident upon increasing the duration of incubation from 24 h to 48 h following ID injection of Cy5 labeled mRNA and eGFP expressing mRNA: longer incubation period clearly favored eGFP expression but facilitated elimination of Cy5 labeled mRNA. Use of the Dermojet^®^ 1-head device and Dermojet^®^ 3-head device resulted in similar eGFP expression, although the Cy5labeled mRNA appeared to be delivered more efficiently with the Dermojet^®^ 3-head device.

**Table 1 pharmaceutics-14-00151-t001:** Summary of various delivery systems, mRNA formulations, and post-delivery evaluations.

**mRNA**	Cy5 Labeled mRNA/eGFP Expressing mRNA		
**mRNA transfection agents**	Liposomal/Polymeric		
**Delivery** **conditions**	**System**	**[Cy5 mRNA]**	**[eGFP expressing mRNA]**
	ID injection	1 µg/30 µL 3 µg/30 µL	1 µg/30 µL 3 µg/30 µL
	Hollow microneedle: MicronJet™600	1 µg/100 µL	1 µg/100 µL
	Jet injector: Dermojet^®^ (1- and 3-nozzle)	1 µg/100 µL	1 µg/100 µL
	Fractional laser ablation: Er:YAG (P.L.E.A.SE.) *	3 µg/30 µL	3 µg/30 µL
**Evaluations**	Biodistribution of Cy5 mRNA visualized by CLSM		
	Biodistribution of expressed eGFP visualized by CLSM		

* Fractional laser ablation was performed on excised skin in Franz cells with conditions (35.4 J/cm^2^ (1.2 W 175 µs; 200 Hz, 3 pulses per pore, density 7%, Ø 10 mm)).

**Table 2 pharmaceutics-14-00151-t002:** Site of mRNA delivery and extent of eGFP expression by the different skin cells.

Delivery Technique	Delivery Site	Types of Cells Expressing eGFP			
		Keratinocytes	Fibroblasts	Vascular Endothelium	Appendageal Epithelium
Intradermal injection	Dermis	+++	+	++	+++
Hollow microneedle injection (MicronJet™600)	Dermis	+++	+	+	+++
Jet injector	Dermis	+++	+	+	++
Fractional laser ablation	Epidermis	+	NA	++	NA

## Data Availability

High resolution images are available in Appendix A.

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
