# Peer review of "Effect of mRNA Delivery Modality and Formulation on Cutaneous mRNA Distribution and Downstream eGFP Expression"

_pharmaceutics, 2022, doi:10.3390/pharmaceutics14010151_

Round 1

Reviewer 1 Report

The authors present robust methods for the visualization of mRNA delivery and subsequent protein expression, and these methods were applied for a systematic comparison of different delivery techniques including invasive methods (conventional intradermal injection and hollow microneedle injection), “less-invasive” methods (jet injection system) and minimally-invasive methods (fractional laser ablation) in this manuscript. It is a very interesting and valuable work. The manuscript is well-structured and suitable for publication in Pharmaceutics.

Author Response

We thank the Reviewer for the time and effort to read our work and highly appreciate the feedback.

Reviewer 2 Report

In my opinion, the paper is intersting and well written but would benefit from additional experiments.

Authors should discuss mRNA routes (intercellular, paracellular or through glands).

Authors need to evaluate the toxicity of each delivery method and quantify ROS.

Authors should evaluate the influence of the delivery methods on tight junctions and skin hydration.

Reviewer 3 Report

In this paper, authors addressed intradermal mRNA injection using porcine skin, and compared four different injection methods: intradermal injection, microneedle injection, jet injection and fractional laser ablation. Distribution of mRNA was studied using Cy5 mRNA and that of protein expression from mRNA was studied using GFP. Although the focus is interesting, results are not convincing especially for GFP because of high level of background. Distribution pattern of GFP fluorescence largely fluctuated. Adjacent sections showed large difference, which cannot explained by mRNA distribution. Indeed Cy5 distribution pattern is not correlated with that of GFP. These data are hard to explain. Each group seems to have n = 1. Is there any rationale to distinguish between background and real fluorescence? Because of low quality of images, quantification is not convincing. It is better to perform ELISA instead for quantification. 

Cy5 distribution pattern is convincing. Thus, all data should be obtained using mRNA expressing red fluorescence or histological staining of GFP with red fluorescent secondary antibody.

Other than this, especially introduction is lengthy containing irrelevant topics. 

I do not recommend to accept this manuscript.

Round 2

Reviewer 2 Report

Authors answered to all my comments. Accordingly I recommend acceptance of the revised paper.

Author Response

We thank the reviewer for having accepted the revised manuscript.

Reviewer 3 Report

I disagree with the comments from authors. 

"It appears clear the GFP expression is not done by random cells

(Table 2: Site of mRNA delivery and extent of eGFP expression by the different skin cells .) – bringing evidence that the GFP signal is not due to background noise."   Non random distribution cannot be evidence for actual signals. More robust evidence is needed.   The authors hypotheses could be true. But no evidence was there. In some cases, signals is high in slide 1, low in slide2, middle in slide 3.... Such strange observation could be explained by uneven distribution of GFP expression but more likely by background fluorescence. Background fluorescence is largely affected by slight change in thickness of section. Whatever the cases, this is quite uncertain. This issue can be solved just by choosing red fluorescence, which is a severe problem in study design.   The answers also lack the convincing explanation about the discrepancy between distribution of Cy5 and EGFP.   These results shown here are too strange, and thus more robust evidences are needed to make them convincing.   Along with this issue, all experiment was done with N = 1 without robust quantification. Thus the results and discussion in this paper is quite uncertain, and thus this paper is not suited as a paper in experimental science field, which should be based on robust evidences.   I disagree on the publication of this manuscript, but let the editors to decide.      
